# Interactions Between Iron Metabolism and the Endocannabinoid System in Bacterial Infections

**DOI:** 10.3390/antibiotics14060614

**Published:** 2025-06-18

**Authors:** Kayle Brenna Dickson, Juan Zhou, Christian Lehmann

**Affiliations:** 1Department of Microbiology and Immunology, Dalhousie University, Halifax, NS B3H 4R2, Canada; kayle.dickson@dal.ca; 2Department of Anesthesiology, Pain Management and Perioperative Medicine, Dalhousie University, Halifax, NS B3H 4R2, Canada; juan.zhou@dal.ca; 3Department of Pharmacology, Dalhousie University, Halifax, NS B3H 4R2, Canada; 4Department of Physiology and Biophysics, Dalhousie University, Halifax, NS B3H 4R2, Canada

**Keywords:** iron metabolism, endocannabinoid system, homeostasis, iron chelation

## Abstract

Iron is a key nutritional requirement for a variety of physiological functions, and its metabolism is tightly controlled under homeostatic conditions. The endocannabinoid system (ECS) represents an additional physiological system with a key role in maintaining homeostasis that is known for its role in modulating immune responses. Recent research has highlighted intriguing interactions between these systems, including the suppression of iron uptake by the ECS and alterations to the iron-catalyzed Fenton reaction. These interactions are particularly interesting in the context of bacterial infections. As iron is a vital nutrient for bacteria, modulating iron levels using the ECS may be able to control bacterial growth. This review aims to explore the current understanding of how the ECS affects iron homeostasis and its implications for bacterial pathogenesis. In this study, we provide an overview of both iron metabolism and the ECS, focusing on harnessing these systems to develop novel therapeutic strategies to modulate iron metabolism in bacterial infections. By elucidating these complex interactions, we hope to provide new insights into the development of novel treatments for bacterial infections.

## 1. Introduction

Antimicrobial resistance (AMR) represents one of the most pressing threats to global health in modern times. In 2019, AMR was estimated to play a role in over 4.95 million deaths, with low- to middle-income countries often being disproportionately impacted [1]. As indicated by a recent metagenomic analysis of wastewater samples from over 60 countries, despite differences in both the abundance and diversity of antibiotic-resistant pathogens across the globe, there remains a high degree of correlation with socioeconomic, health, and environmental factors [2]. Though evidence from low- and middle-income countries remains limited, current estimates suggest that resistant infections are associated with substantial increases in the cost of medical care [3]. This burden is expected to increase significantly, as it is estimated that, by 2050, up to 8.22 million deaths may be associated with AMR [4].

The persistent threat of AMR has spurred a number of significant initiatives at the global and national level. As reviewed by Mudenda et al., current efforts involve the creation of strong surveillance programs, the implementation of vaccination and sanitation programs, and the development of both new antibiotics and alternative therapies [5]. Historically, combination therapies have been employed to combat antibiotic-resistant pathogens, but this strategy may add evolutionary pressure to pathogens, resulting in the emergence of multidrug-resistant pathogens [6]. As of 2024, the World Health Organization (WHO) has 24 pathogens, representing 15 distinct families, on its Bacterial Priority Pathogens List [7]. Taken in combination with the challenges associated with administering combination therapies, this has placed additional importance on the development of novel therapeutic options.

The discovery of antibiotic compounds, dating back to the late 19th century, heralded an era of rapid scientific development. As reviewed by Nicolaou and Rigol, the antibiotic pipeline remained strong for decades, as new antibiotic compounds were identified or synthesized in the laboratory [8]. Chemical modifications further enhanced the number of antibiotics available, as existing compounds were altered to enhance their efficacy or reduce associated side effects. Compared with the rapid pace of development seen during the 20th century, the current antibiotic pipeline is stagnant. The WHO conducts annual reviews on the state of the antibiotic pipeline, and has reported that only 16 new anti-bacterial compounds have been approved for clinical use since 2017 [9]. This lack of development can be partially attributed to the changing landscape of drug development, including the limited returns expected on investments in R&D, and a variety of regulatory issues, such as increasingly stringent requirements for clinical trials.

An additional problem facing the antibiotic pipeline is the relative lack of new bacterial targets, which limits our ability to develop antibiotics with new modes of action. Though agents against new bacterial targets (e.g., quorum sensing, biofilm synthesis, etc.) are in pre-clinical development, alternative approaches to antibiotic discovery and development are now of particular interest to the research community [10]. A recent review by Czaplewski et al. identified 19 alternative approaches to antibiotics, which were classified into tiers based on their development stage [11]. Two of the listed approaches are relevant to this review: metal chelation and immune modulation. These approaches will be detailed in detail in the following sections.

The introduction of the first siderophore cephalosporin, cefiderocol (Fetroja^®^), has opened a new avenue to overcoming AMR, disrupting bacterial iron metabolism while also supporting iron-related pathways in inflammation and infection in the human host [12]. The iron levels in humans are tightly controlled via a complex network of regulating mechanisms. One of these participating systems is the endocannabinoid system (ECS). The aim of this review is to explore interactions and potential synergies between iron metabolism and the ECS. When combined, these two complementary strategies may represent a viable solution to the current challenges associated with the management of infections.

## 2. Antibiotic Effects of Iron Chelators

Iron is a key metabolite required by humans for a variety of physiological processes, such as oxygen transportation, DNA synthesis, etc. Despite the importance of iron to human physiology, free iron can engage in redox chemistry, generating cytotoxic reactive oxygen species. As such, iron metabolism must be tightly regulated to ensure availability for essential processes while preventing high levels of free intracellular iron. As humans require quantities of iron for erythropoiesis that exceed typical daily intake levels, we rely primarily on macrophage phagocytosis for recycling iron within senescent red blood cells. Hepcidin coordinates this response by altering levels of the iron exporter ferroportin within macrophages and other iron storage sites (i.e., hepatocytes) [13]. Bacteria also require tight control of iron homeostasis to allow for bacterial growth while avoiding oxidative damage. In contrast to human iron metabolism, bacteria rely on a variety of iron uptake systems and secreted siderophores to maintain intracellular iron stores [14]. Many bacterial species produce siderophores, which are small chelator-like molecules that are released into the environment where they act as iron-scavenging systems.

Iron chelators are small molecules that can bind to iron, making it inaccessible for cellular processes. These molecules are structurally similar to bacterial siderophores and are able to tightly sequester iron away from the host system. Several iron chelators, such as deferiprone (DFP), deferoxamine (DFO), and deferasirox (DFX), are approved by the United States Food and Drug Administration and as such are available for clinical use in the treatment of iron overload disorders [15]. More recently, iron chelation has been proposed as a novel treatment for infections [16].

The human body naturally employs iron restriction during infection, known as nutritional immunity, to limit the iron available for bacterial metabolism [17]. This is a dynamic process that results in the internalization of the iron exporter ferroportin in response to the production of the iron-regulator protein, hepcidin. Inflammation, specifically lipopolysaccharide-mediated TLR-4 signaling and IL-6 secretion, further upregulates the production of hepcidin by hepatocytes to reduce levels of bioavailable iron [17]. Appropriately designed iron chelators can support nutritional immunity by acting as a “sink” for free iron, effectively sequestering this iron and lowing systemic iron levels. This results in a decrease in the growth of the iron-deprived bacteria, with the chelator essentially acting as a bacteriostatic antibiotic. An overview of some of the pathogens that iron chelators have been determined to have anti-bacterial effects against is provided in Table 1. The chemical structures of the iron chelators that are currently approved for clinical use are presented in Figure 1.

If used clinically for the treatment of infections, the approved small molecule iron chelators (e.g., DFO, DFP, and DFX) have several limitations that must be considered, namely relating to the route of administration (e.g., parenteral vs. oral), high doses required, and the potential for serious side effects. Siderophore-based DFP has been linked to an increased risk of severe infections, relating to the development of neutropenia [29]. DFP and DFX also require a relatively high dose for the treatment of infections, which limits their usability in this setting [30]. Additionally, some small chelators can be used as siderophores by bacteria, placing patients at an increased risk of infection. For example, DFO is derived from a siderophore produced by *Streptomyces pilosus*; this limits its clinical utility as it can be taken up by invading pathogens and used as a source of iron for bacterial metabolism [31]. Additionally, DFO can enhance the growth of *K. pneumoniae* in vitro, and has been shown to increase bacterial loads in mice infected with a community-acquired strain of methicillin-resistant *S. aureus* [32,33]. This has spurred the development of additional chelators that are able to outcompete bacterial iron acquisition.

One strategy for overcoming bacterial iron acquisition systems is the development of iron chelators with enhanced iron binding capabilities. DIBI is an iron-binding polymer that is highly selective for iron, highly bioavailable (ip/iv), water soluble, and non-toxic in murine models [23,34,35]. Based on synthetic hydroxypyridinone iron chelators, DIBI consists of nine 3-hydroxy-1-(β-methacrylamidoethyl)-2-methyl-4(1H)-pyridinone (MAHMP) residues on a polyvinylpyrrolidone (polymer backbone), which allows for the coordination of three molecules of iron per polymer [23]. This polymer backbone has the additional advantage of inducing a wrapping effect around the sequestered iron, protecting the iron from external influences [36]. The enhanced size of the chelator also prevents uptake by bacteria. In terms of its antimicrobial effects, DIBI is synergistic with aminoglycosides, fluoroquinolones, and glycopeptides (gentamicin, ciprofloxacin, and vancomycin, respectively) and is more inhibitory to *S. aureus*, *A. baumannii* and *C. albicans* than either DFP or DFO in vitro [22]. In vivo, DIBI has shown promising effects for the treatment of MRSA, polymicrobial sepsis, and lung injury induced by *P. aeruginosa* [22,37,38]

One additional strategy that has been employed to overcome the chelator uptake that is seen with small molecules such as DFP is to design chelators with a higher molecular weight. Once molecules exceed a weight of 600 Da, they cannot be readily taken up by bacteria, preventing usage of the iron contained within. One such example is the polyglycerol-based family of macromolecular iron chelators, which consist of hyperbranched polyglycerol (HPG) in combination with the high-affinity hexadentate chelator, *N*,*N*-bis (2-hydroxybenzyl) ethylenediamine-*N*,*N*-diacetic acid (HBED) [18]. These chelators have bacteriostatic activity against *S. aureus* in vitro and, when combined with rifampicin, against MRSA. This suggests that these chelators may have utility in the management of infections, reducing the need for high doses of antibiotics. Similarly, new chelators have been developed that suggest that, when conjugated to hyperbranched polyglycerol, DFO exhibits improvements in its half-life, while still efficiently sequestering iron [39,40]. These chelators are yet to be evaluated for antimicrobial effects, but evidence suggests that additional macromolecular chelators, including those consisting of 3-hydroxypyridin-4-ones (HPOs) conjugated to poly (glycidyl methacrylate) (PGMA), have activity against MRSA.

Siderophore–antibiotic conjugates represent an additional strategy that allows for iron chelation to be used in the management of infections. These conjugates are based on sideromycins, such as albomycin, which are naturally produced antibiotics that mimic bacterial siderophores [41]. As reviewed by Möllmann et al., sideromycins essentially act as a vehicle for the transport of compounds into bacteria, exploiting their iron acquisition pathways. Based on this mechanism of action, siderophore–antibiotic conjugates have been referred to as a “Trojan horse.” As one example, siderophore–aminopenicillin conjugates are actively taken up by *P. aeruginosa*, resulting in good anti-bacterial activity in vitro and in vivo [42]. The siderophore-conjugated cephalosporin cefiderocol is highly effective against multi-drug-resistant strains and is able to withstand beta-lactamase activity, though the mechanisms behind this remain unclear [43]. Importantly, cefiderocol is currently in phase III clinical trials, which are investigating its efficacy against serious infections caused by carbapenem-resistant Gram-negative pathogens (CREDIBLE-CR; NCT02714595) [44].

## 3. Antibiotic Effects of Cannabinoids

The endocannabinoid system (ECS) is an endogenous lipid-based signaling system that plays a key regulatory role in many physiological processes. As reviewed by Chanda et al., the ECS has a well-described role in maintaining homeostasis within the central and peripheral nervous systems, and regulating metabolism in a variety of tissues [45]. Additionally, the ECS can modulate many different pathophysiological conditions, including but not limited to chronic stress, obesity, neurological diseases, inflammatory/autoimmune conditions, and cancers.

The ECS is primarily composed of cannabinoid receptors 1 (CB1R) and 2 (CB2R). Historically, CB1R was considered to be the central cannabinoid receptor, located primarily within the central nervous system, while CB2R was thought to be primarily localized to the peripheral tissues, with a more restricted pattern of expression within the brain [46,47]. Beyond the CNS, CB2R is predominantly found on immune cells, with high levels of expression on natural killer cells, B cells, and monocytes and lower levels of expression on T cells and neutrophils [47,48]. A simplified schematic of receptor distribution within the ECS is shown in Figure 2A. Also included in the ECS are the endogenous ligands arachidonoylethanolamide (AEA) and 2-arachidonoyl glycerol (2-AG), and associated enzymes. These lipid-based molecules are derived from arachidonic acid, and are synthesized in response to a variety of physiological signals, including changes in energy consumption, stress levels, mood, pain, and inflammation [49]. They exert their effects as intercellular messengers in either a paracrine or autocrine manner. Phytocannabinoids and synthetic cannabinoids represent additional classes of exogenous cannabinoid molecules. These exogenous ligands can be found in variety of plants, including, but not limited to, *Cannabis sativa*. The various signaling pathways activated by either endogenous or exogenous cannabinoids can have a variety of downstream effects. In brief, cannabinoid signaling pathways result in changes in gene transcription, protein translation, metabolism, cell growth/survival, and other additional cellular functions [50,51]. The key components of the ECS, as relevant to this manuscript, are presented in Figure 2B.

The potential role of the ECS in infection is a relatively new subject within the field, which has historically focused on the ability of cannabinoids to modulate the immune response to infection. Cannabinoids are often broadly considered immunosuppressive, but the endocannabinoid system appears to have context-specific effects. Though AEA largely exerts anti-inflammatory effects, 2-AG appears to induce both pro- and anti-inflammatory effects depending on the context [52]. For example, 2-AG can promote vascular inflammation by enhancing leukocyte recruitment in the context of atherosclerosis [53]. With respect to exogenous cannabinoids, Maggirwar and Khalsa summarize the links between cannabis use (i.e., exogenous cannabinoids) and impaired immune responses, focusing on the impact on anti-viral immunity and respiratory infections [54]. Cannabis use has also recently been associated with an anti-inflammatory T cell phenotype in individuals with HIV, with reductions in the abundance of effector T cells and the expression of activation markers [55]. Despite this phenotype, HIV-specific T cell responses were preserved. Similar immune suppression has been observed in the case of bacterial infections, with cannabis use being associated with suppressed immune responses to oral bacterial pathogens [56]. These and other data suggest caution is warranted when modulating the endocannabinoid system in the context of infection, particularly in vulnerable populations, as cannabinoid-mediated immune suppression can pose challenges for clearance of infection.

Despite these concerns, cannabinoids can be harnessed in other ways, often in conjunction with antibiotics to combat infections. To this end, cannabinoids have been reported to have direct anti-bacterial, anti-viral and anti-fungal effects, and may also have indirect effects that contribute to the resolution of infection.

The earliest studies on the anti-bacterial effects of cannabinoids investigated the effects of extracts, which contain a wide range of compounds. *Cannabis sativa* extract was found to have anti-bacterial effects against both Gram-positive (e.g., *Bacillus* spp., *S. aureus*, and *Micrococcus flavus*) and Gram-negative (e.g., *Proteus vulgaris* and *Bordetella bronchioseptica*) species, as well as anti-fungal effects against *Aspergillus niger* [57]. Of note, many cannabinoid compounds are effective against several highly resistant pathogens, including methicillin-resistant *S. aureus* (MRSA) [58]. With respect to viral infections, some evidence suggests that both Δ^9^-tetrahydrocannabinol (THC) and cannabidiol (CBD) can block viral translation and replication of the main protease of SARS-CoV-2 in vitro [59]. Despite this promising data, these effects are often challenging to translate into in vivo models, as the required dose of cannabinoids often exceeds practical therapeutic ranges. As such, this suggests that indirect anti-microbial effects may be of more importance.

Cannabinoids can exert several direct and indirect effects against MRSA, a Gram-positive pathogen that is able to form biofilms on both necrotic tissue and medical devices (i.e., titanium implants). Biofilms confer a significant advantage to these bacteria in terms of resistance to both antibiotics and the immune response. Farha et al. demonstrated that in vivo, cannabigerol both inhibits MRSA biofilm formation and breaks down existing biofilms, and inhibits the growth of MRSA at a concentration of 2 μg/mL [60]. The authors identified cannabigerol as a membrane perturbant, affecting the cytoplasmic membrane of Gram-positive pathogens, and determined that this activity could also extend to Gram-negative species when administered with Polymyxin B to permeabilize the outer membrane and allow access to the cytoplasmic membrane.

Additional non-cannabinoid compounds can also interact with the ECS to produce anti-bacterial effects. Terpene beta-caryophyllene (BCP) is one such compound that has activity at CBR2. Several studies have demonstrated the anti-bacterial effects of BCP with respect to both Gram-positive and Gram-negative pathogens. An early study by Schmidt et al. showed good anti-bacterial efficacy against Gram-positive pathogens (i.e., *Staphylococcus aureus* and *Enterococcus faecalis*), and moderate efficacy against Gram-negative pathogens (i.e., *E. coli*, *Pseudomonas aeruginosa*, *Proteus vulgaris*, *Klebsiella pneumoniae*, and *Salmonella abony*) [61]. Time–kill curve studies suggest that *S. aureus* killing occurs rapidly, within four hours of BCP administration [62]. BCP is also effective against species involved in dental plaque formation in dogs, with the inhibition of Streptococcus species being of particular importance [63]. Additionally, sub-inhibitory concentrations of BCP impaired biofilm formation for a key contributor to early plaque formation: *Streptococcus mutans*. While the mechanisms behind these effects are largely unknown, one study implicates altered membrane permeability, leading to leakage of intracellular contents, as the mechanism behind the bactericidal effects of BCP on *Bacillus cereus* [64].

Indirect effects of cannabinoids have also been observed against several viruses, including Hepatitis C (HCV) and SARS-CoV-2. For HCV, CBD has no direct effects on viral replication, but is cytotoxic against the hepatic cells used to culture the virus [65]. CBD can also suppress the proliferation of T cells and macrophages, which are known to mediate damaging inflammatory processes within the liver. Taken together, this evidence suggests that CBD may be used to manage HCV. With respect to SARS-CoV-2, essential oils derived from hemp, containing high levels of the terpenes trans-caryophyllene and α-pinene, inhibited SARS-CoV-2 infections by decreasing the expression of angiotensin-converting enzyme 2 and transmembrane protease serine 2 [66]. These molecules serve as key receptors for viral entry within the lung.

## 4. Potential Interactions and Synergies

Interactions between iron metabolism and the ECS have been described in various conditions, including those associated with an iron overload. Kappel da Silva et al. described the neuroprotective potential of CBD in a murine model of cognitive decline triggered by an iron overload within the brain [67]. Dysregulated iron metabolism in patients with inflammatory bowel disease has been linked to increased levels of intracellular iron, causing osteoclast hyperactivation and bone resorption; CB2R activation can reduce this hyperactivity, and may represent a therapeutic target for osteoporosis related to inflammatory bowel disease [68]. Cannabinoids have also been shown to interact with iron metabolism without directly acting on cannabinoid receptors. Cannabinol has been shown to be neuroprotective, inhibiting the oxytosis/ferroptosis cell death pathway and protecting against mitochondrial dysfunction [69]. These selected examples highlight the breadth of known interactions between these two systems.

Evidence also supports potential interactions between iron metabolism and the ECS that may play a role in mediating bacterial infections. The divalent metal transporter-1 (DMT1) mediates dietary iron uptake across the intestinal mucosa and facilitates the peripheral delivery of iron released by transferrin in the endosome. Seo et al. reported that classical cannabinoids (THC), nonclassical cannabinoids (CP 55,940), aminoalkylindoles (WIN 55,212-2), and endocannabinoids (AEA) reduce iron uptake by HEK293T cells stably expressing DMT1 [70]. siRNA knockdown of CB2R abrogate this inhibition. Since anti-inflammatory actions mediated through CB2R would be associated with reduced DMT1 phosphorylation, the authors postulate that this pathway provides a means to reduce oxidative stress by limiting iron uptake. For bacterial infections, this could potentially translate to reduced bacterial killing due to a decrease in the generation of reactive oxygen species (ROS) by leukocyte subpopulations. On the other hand, iron starvation of the organism also limits bacterial growth. A recent study by Creoli et al. confirmed the relationship between DMT1 expression, which was increased, and the ECS in pediatric patients with inflammatory bowel disease [71]. CB2R activation restored iron metabolism by downregulating DMT1 expression. This pathway merits further research to characterize the effects of DMT1 downregulation on immune function and the subsequent control of bacterial infections. Control of iron levels during infection is likely to be the most effective when levels of free iron remain low, but iron is still available to host cells for executing critical immune functions.

Another intriguing crosslink between iron metabolism and cannabinoids is represented by the effects of cannabinoids on oxidative stress and the Fenton reaction, which is catalyzed by iron. As reviewed by Gallelli et al., significant cross-talk occurs between the ECS and oxidative stress; CB1R and CB2R activation can have differential effects on oxidative stress, with CB1R agonism increasing the activity of ROS-generating enzymes, while CB2R decreases this activity [72]. These effects are likely to be disease- and tissue-specific, and could be amplified by changes in iron metabolism. THC has also been shown to alter oxidative stress by blocking the Fenton Reaction, thus preventing the increase in ROS production that is normally induced by high intracellular iron levels. Carter et al. showed in bone marrow-derived cells (BMDCs) that THC exposure resulted in decreased intracellular ROS production, both by blocking the formation of ROS through the Fenton reaction and inducing the antioxidant defense system Nrf2-ARE within the cell, ultimately blocking the differentiation of BMDCs into macrophages [73]. This effect of THC was independent of cannabinoid receptors CB1 and CB2R, as well as other potential receptors, such as GPR18, GPR55, and Adenosine 2A, representing a unique molecular property associated with THC that could affect the immune response to bacterial infections. CBD has also been shown to regulate oxidative stress by promoting the Nrf2-ARE pathway and chelating ferrous iron [74]. CBD also has well-described anti-bacterial effects, which position CBD as an optimal candidate for investigating the ECS and iron metabolism in the context of bacterial infections [75].

With regard to ferroptosis modulation, the CB2R agonist, beta-caryophyllene (BCP), has been shown to inhibit this pathway, suggesting that BCP is a promising therapeutic strategy for treating ferroptosis-associated disorders [76]. Sepsis is one such example, as ferroptosis is associated with organ damage [77]. However, in a bacterial infection, ferroptosis is considered a “mixed blessing” (Figure 3 adapted from [78]): lipid peroxidation induces plasma membrane destabilization, leading to ferroptosis-mediated cell death with *Mycobacterium tuberculosis* (Mtb) infection, driving macrophage necrosis and allowing *M. tuberculosis* to thrive and spread, which promotes the infection [79]. On the other hand, ferrous iron could be delivered to the intracellular bacterial vacuole, eventually inducing the ferroptosis-like death of bacteria, which assists in the killing of bacteria (e.g., *E. coli* or *S. aureus*) by macrophages [80]. This suggests that the potential benefits of ferroptosis modulation in bacterial infections are likely to be disease- and pathogen-specific. In the context of sepsis, modulation of ferroptosis would require knowledge on the causative agent to avoid macrophage necrosis, leading to further dissemination of the pathogen and further impairing immune function.

## 5. Conclusions

Both iron and the ECS play important roles in the host response to bacterial infection, with iron restriction forming a key component of nutritional immunity, and the ECS playing a role in the modulation of immune responses and inflammation. Though both systems are involved in bacterial infections, they offer distinct but complementary pathways for antimicrobial control. Current evidence suggests that these systems interact in a variety of ways, suggesting possible synergy that may be of benefit therapeutically. Despite these possible benefits, additional research is required to determine the exact clinical scenarios in which dual modulation of the ECS and iron metabolism would have utility. Future research should directly investigate the interactions between iron and the ECS, with the hope of developing novel adjunct therapeutics for bacterial infections.

## Figures and Tables

**Figure 1 antibiotics-14-00614-f001:**
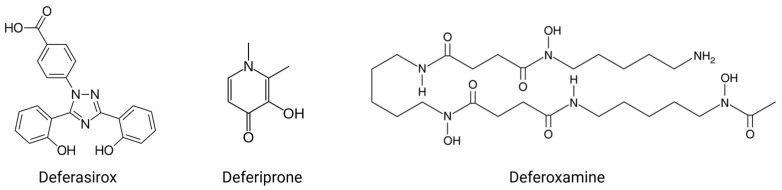
Chemical structures of clinically approved iron chelators. Figure created on Biorender.com.

**Figure 2 antibiotics-14-00614-f002:**
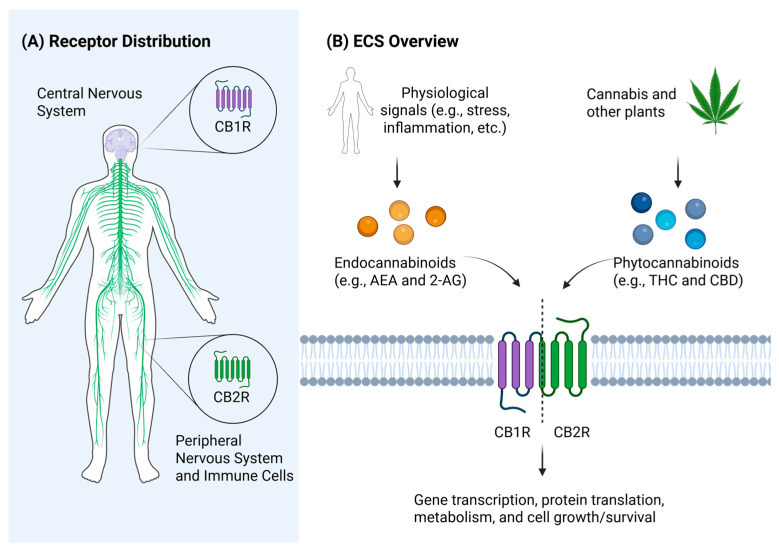
(**A**) Simplified representation of cannabinoid receptor distribution. CB1R is primarily found within the CNS, while CB2R is located in the periphery and on immune cells. (**B**) Overview of the ECS. Endocannabinoids AEA and 2-AG, or phytocannabinoids (e.g., THC and CBD) act on CB1R or CB2R depending on the molecule of interest. Receptor agonism results in changes to gene transcription, protein translation, metabolism, and cell growth/survival, among other actions. Figure created on Biorender.com.

**Figure 3 antibiotics-14-00614-f003:**
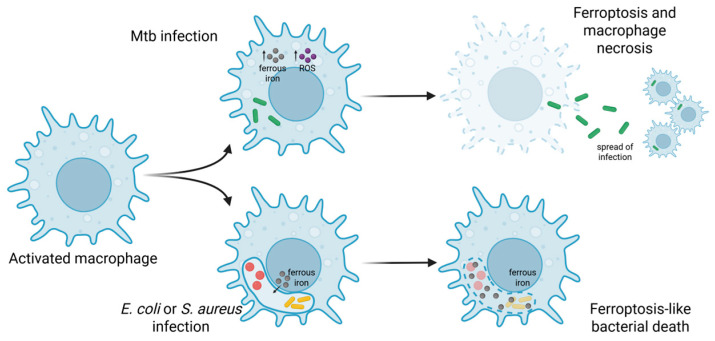
Ferroptosis and bacterial infection. In Mtb infection, increased levels of ferrous iron and ROS lead to membrane breakdown, resulting ferroptosis and macrophage necrosis (upper scenario). This allows Mtb to disseminate to other cells. In other infections (e.g., *E. coli* or *S. aureus*), bacteria undergo a ferroptosis-like death within the vacuole. Figure adapted from [78] and created with Biorender.com. Mtb—*Mycobacterium tuberculosis*.

**Table 1 antibiotics-14-00614-t001:** Iron chelators and associated antibiotic effects. Chelators are listed according to developmental stage (pre-clinical development or approved for clinical use), along with target pathogens and associated references.

Stage	Iron Chelator	Target Pathogens	References
Pre-clinical	MRB20	*Staphylococcus aureus*, *Staphylococcus epidermidis*, *Enterococcus faecium*, *Enterococcus faecalis*	[18]
	VK28	*Acinetobacter baumannii*, *S. aureus*	[19]
	Apo6619	*A. baumannii*, *Escherichia coli*	[19]
	ApoL1	*A. baumannii*, *Klebsiella pneumoniae*, *Pseudomonas aeruginosa*	[19]
	Hexadentate hydroxypyridinone **7**	*S. aureus*, *Providencia stuartii*	[20]
	HPO–PGMA polymers	*Methicillin-Resistant S. aureus* (MRSA)	[21]
	DIBI	*S. aureus*, *A. baumannii*	[22,23]
	HPG–HBED-based macrochelators	*S. aureus*, MRSA	[18]
	Open merocyanine isomer	*E. coli*	[24]
	CP762	*P. aeruginosa*	[25]
Clinical (iron overload)	Deferoxamine	*Prevotella intermedia*, *E. coli*, *S. aureus*, *P. aeruginosa*	[26,27]
	Deferasirox	*P. intermedia*, *E. coli*, *S. aureus*, *P. aeruginosa*	[26,27]
	Deferiprone	*S. epidermidis*, *E. coli*, *S. aureus*, *P. aeruginosa*, *A. baumannii*, *K. pneumoniae*	[19,27,28]

## Data Availability

No new data were created or analyzed in this study.

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
