# Peer review of "Interactions Between Iron Metabolism and the Endocannabinoid System in Bacterial Infections"

_antibiotics, 2025, doi:10.3390/antibiotics14060614_

Round 1

Reviewer 1 Report

Comments and Suggestions for Authors

This manuscript is overall well-written and easy to understand. I enjoyed reading it. The authors have identified an interesting and novel area of research by exploring the intersection of two distinct physiological systems: iron metabolism and the endocannabinoid system (ECS) in the context of bacterial infections.

The review successfully details recent advancements in using iron chelation and ECS modulation as independently viable approaches to combatting infection. However, the central thesis concerning the synergy between these two systems could be developed more thoroughly. I recommend acceptance with minor revisions with the following comments:

Major comments:

  1. The discussions on iron chelation (section 2) and the ECS (section 3) are excellent and detailed. However, the section on their interactions and potential synergy (section 4) is limited in comparison. To better align with the title and the stated aim of exploring "potential synergies", it would be beneficial for the authors to expand this section. A more detailed explanation of how this synergy might applied therapeutically would significantly strengthen the manuscript. Alternatively, the authors could consider changing the title to pivot away from the interactions between the two systems and focus more on them as separate alternative therapies.
  2. The manuscript currently contains only one figure. To reach for a broader audience, the authors are encouraged to include more illustrative figures. Suggestions include:
    • A figure for section 2: illustrating the chemical structures of promising iron chelators and their mechanism of action (such as "Trojan horse")
    • A figure for section 3: summarizing the key components of the ECS (receptors, ligands) and their role in modulating immune responses.
    • A figure for section 4: visually integrates the two systems, showing the specific points of interaction discussed

Minor comments:

  1. Line 83 - add citation
  2. A few acronyms are introduced without their full name being provided upon first use. Please define the following:

Line 88: LPS-mediated TLR-4

Line 189: THC, CBD

Reviewer 2 Report

Comments and Suggestions for Authors

This review is engaging, timely, and thoughtfully written. It effectively integrates two distinct yet interconnected biological systems—iron metabolism and the endocannabinoid system (ECS)—to explore their potential interplay as novel avenues for combating bacterial infections. The authors skillfully combine fundamental science, mechanistic insights, and potential therapeutic applications. This multidisciplinary approach is particularly valuable and relevant given the urgent global challenge posed by antimicrobial resistance.

I have a few suggestions to further enhance the manuscript:

  1. Clarify the Endogenous Nature of the ECS: While the paper discusses cannabinoid compounds and their antimicrobial effects, it would benefit from explicitly distinguishing the ECS as an endogenous regulatory system active within the human body, independent of exogenous cannabinoid intake. Endocannabinoids are naturally synthesized in response to physiological stress, immune signaling, and homeostatic demands. Emphasizing this distinction will help readers appreciate that the ECS is not solely activated by cannabis-derived compounds. Additionally, consider briefly mentioning indirect strategies to support or modulate the ECS, such as exercise and stress management would provide a broader context for potential ECS-targeted interventions.
  2. Differentiate Endocannabinoids from Exogenous Cannabinoids with Antimicrobial Activity: The review attributes antimicrobial properties primarily to exogenous cannabinoids such as THC, CBD, and cannabigerol. It should clarify that these activities pertain to plant-derived or synthetic cannabinoids, and are distinct from the functions of endogenously produced endocannabinoids. Moreover, discussing challenges related to cannabinoid-based therapies, such as dose dependency, psychoactive side effects, and potential immune suppression risks, especially in vulnerable patient populations would provide a balanced perspective.
  3. Table 1: It is unclear why CP762 is separated from other preclinical candidates or whether it is preclinical or clinical.  Furthermore, the alignment of chelator names with their respective target pathogens is somewhat ambiguous. I suggest improving formatting by adding grid lines to reduce potential misinterpretation.
  4. Line 125: Please provide the names and classes of antibiotics reported to have synergistic effects when combined with DIBI to improve specificity and utility.
  5. Line 92: Rephrase for Clarity, “For example, DFO is derived from a siderophore that is naturally produced by Streptomyces pilosus, which limits its utility as it can be taken up by invading pathogens and used as an iron source”

Reviewer 3 Report

Comments and Suggestions for Authors

In this review, Dickson et al. discuss the importance of targeting iron metabolism and the emerging roles of cannabinoids, as well as the potential crosslink between them in combating pathogenic infections. While the review provides something interesting for readers about a novel mechanism controlling infections, I made some suggestions to improve the manuscript before it can be considered for publication.

(Comments)

- The authors attempt to introduce the key terms of the paper in the abstract. However, it does not seem to provide a concise general overview. It would be nice to reorganize the content by first providing a general background about iron metabolism and ECS, along with their important roles in controlling bacterial infections.

- Lines 26-27: “In 2019” is repeated in the same sentence. Please revise it for clarity.

- Line 68: Before the opening sentence, a brief discussion of the role of iron metabolism in bacteria and the host would provide helpful context.

- The manuscript would benefit from professional editing to correct typos and grammar issues and to improve overall clarity. Some sentences are incomplete and missing final punctuation marks. (e.g., Line 33, 62-63, 138-140, etc.)

- Some abbreviations were not defined in the main text. For clarity, it would be helpful to define all abbreviations at their first mention.
